

**Overcoming barriers to enable convergence research by integrating ecological**

**and climate sciences: The NCAR-NEON system Version 1**

Danica L. Lombardozzi[1§*], William R. Wieder[1,2,§,*], Negin Sobhani[1], Gordon B. Bonan[1], David Durden[3],
Dawn Lenz[3], Michael SanClements[3], Samantha Weintraub-Leff[3], Edward Ayres[3], Christopher R. Florian[3],
Kyla Dahlin[4], Sanjiv Kumar[5], Abigail L. S. Swann[6], Claire Zarakas[6], Charles Vardeman[7], Valerio
Pascucci[8]
[1] Climate and Global Dynamics Laboratory, National Center for Atmospheric Research, Boulder CO, USA.
[2] Institute of Arctic and Alpine Research, University of Colorado Boulder, Boulder CO, USA.
[3] National Ecological Observatory Network, Battelle. Boulder CO, USA.
[4] Michigan State University, East Lansing MI, USA.
[5] College of Forestry, Wildlife and Environment, Auburn University, Auburn AL, USA.
[6] University of Washington, Seattle WA, USA.
[7] Center for Research Computing, University of Notre Dame, Notre Dame, IN, USA
[8] Scientific Computing and Imaging Institute, University of Utah, Salt Lake City, UT, USA
[§] Contributed equally as lead authors.
[*] Correspondence to: dll@ucar.edu and wwieder@ucar.edu



## Abstract

Global change research demands a convergence among academic disciplines to understand
complex changes in Earth system function. Limitations related to data usability and computing
infrastructure, however, present barriers to effective use of the research tools needed for this cross-
disciplinary collaboration. To address these barriers, we created a computational platform that pairs
meteorological data and site-level ecosystem characterizations from the National Ecological Observatory
Network (NEON) with the Community Terrestrial System Model (CTSM) that is developed with university
partners at the National Center for Atmospheric Research (NCAR). This NCAR-NEON system features a
simplified user interface that facilitates access to and use of NEON observations and NCAR models. We
present preliminary results that compare observed NEON fluxes with CTSM simulations and describe
how the collaboration between NCAR and NEON that can be used by the global change research
community improves both the data and model. Beyond datasets and computing, the NCAR-NEON
system includes tutorials and visualization tools that facilitate interaction with observational and model
datasets and further enable opportunities for teaching and research. By expanding access to data,
models, and computing, cyberinfrastructure tools like the NCAR-NEON system will accelerate integration
across ecology and climate science disciplines to advance understanding in Earth system science and
global change.

## Short Summary

We present a novel cyberinfrastructure system that uses National Ecological Observatory Network
measurements to run Community Terrestrial System Model point simulations in a containerized system.
The simple interface and tutorials expand access to data and models used in Earth system research by
removing technical barriers and facilitating research, educational opportunities, and community
engagement. The NCAR-NEON system enables convergence of climate and ecological sciences.

## 1. Introduction

Earth system science aims to deepen understanding of interactions between natural and social
systems and their responses to global change. As such, the collective understanding of changes in Earth
system function in response to global change drivers requires a convergence among scientific disciplines,
including physical and natural sciences (Kyker-Snowman et al. 2022). This research combines a variety
of complex observational data with ever more sophisticated computational models. Notably, Earth System
Models (ESMs) are essential tools for assessing and predicting our changing environment (Bonan and
Doney 2018), but limitations related to data usability and access to computing infrastructure present
barriers to effective use of these research tools (Fer et al. 2021). Addressing these barriers is critical to
engage the broad, cross-disciplinary communities that are required for Earth system science research,



education, and training (NASEM, 2022). We feel that tractable progress can be made to reduce these
data and technical barriers to better understand and project changes in Earth system function under
global change.

The availability, discoverability, and usability of observational data are essential to running,

calibrating, and validating models. For example, the scientific advancements made in measuring eddy
covariance (EC) fluxes have been critical to the development, evaluation, and improvement of the
representation of terrestrial ecosystems in ESMs. Initially, model-data comparisons were limited to short,
intensive field campaigns extending over a few weeks (Bonan et al. 1997), but this grew to comparison
with flux network datasets extending over several years at multiple sites (Stöckli et al. 2008), and
comparison with globally gridded flux products (Bonan et al. 2011; Jung et al. 2020). Flux tower data sets
continue to provide essential information for land model development and evaluation (Best et al. 2015;
Lawrence et al. 2019). Notably, single-point simulations can use EC measurements to facilitate more
rapid model development and testing of ecological hypotheses (Bonan et al. 2012; Burns et al 2018;
Swenson et al. 2019; Wieder et al. 2017). An explosion of EC measurements and strong network
coordination make these data easier to find (Durden et al. 2020; Pastorello et al. 2020), but the need to
perform additional data processing prior to use presents barriers to integrating ecological observations
into land model development and evaluation. These barriers include gap filling associated meteorological
data, assessing EC flux data quality, and persistent challenges in discovering and harmonizing
complementary data – including information about vegetation and soils at EC tower sites. Our work seeks
to provide a framework to address these data challenges to facilitate the integration of local meteorology,
EC flux measurements, and ecosystem characterizations in the development and evaluation of land
models that are used for Earth system prediction and global change research.

Beyond these data challenges, barriers to accessing and using computing infrastructure also

impede broader community engagement with tools that are central to global change research. This limits
the participation of scientists from environmental science and ecology, which are fundamental
components of the Earth system, in the development and use of ESMs. The Community Earth System
Model (CESM; Hurrell et al. 2013; Danabasoglu et al. 2020) has a long history of being freely and openly
available to users, yet several barriers related to training, cyberinfrastructure, and data integration have
hampered broader adoption and use of this model by a wide range of researchers. Thus, model code
may be publicly available, but access to computing resources and the associated technical expertise
needed to use them presents barriers to engaging a diverse, cross-disciplinary community of model users
who can harness these powerful tools for research and teaching. We contend that broader engagement
across scientific disciplines is critical to improving the representation of Earth system processes and their
likely responses to global change.

This work  overcomes some of the barriers to the use of ESMs in ecology by creating an

integrated 'NCAR-NEON system'. This system combines meteorological data and site-level ecosystem
characterizations from the National Ecological Observatory Network (NEON) with the Community



Terrestrial System Model (CTSM), an extension of the Community Land Model (CLM5; Lawrence et al.
2019). CTSM is the terrestrial component of CESM, which is developed with university partners at the
National Center for Atmospheric Research (NCAR; Fig. 1). The NCAR-NEON system also features a
simplified user interface that facilitates access to and use of NEON observations and NCAR models. By
developing this NCAR-NEON system, we aim to enable the convergence of climate and ecological
sciences by providing accessible cyberinfrastructure, quality-controlled datasets from NEON, and tutorials
for analyzing and visualizing observed and simulated data. We describe development of the NCAR-
NEON system, present results comparing observed NEON fluxes with simulations from CTSM, and
outline opportunities that the system enables for research and education across scientific disciplines.

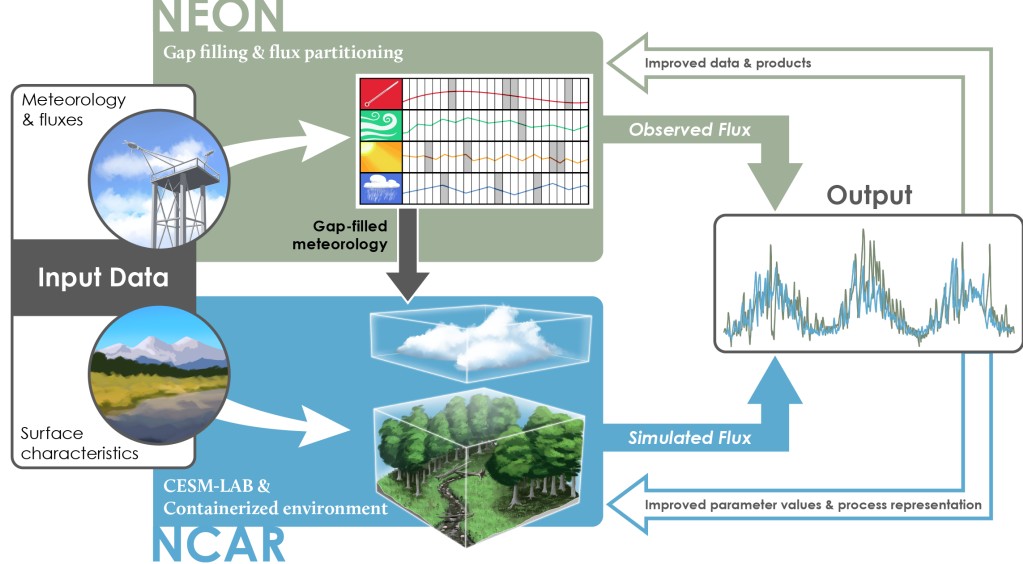


***Figure 1.*** *A conceptual diagram illustrating the integration of NEON data and NCAR modeling enabled through the*
*NCAR-NEON system. NEON meteorological measurements are gap-filled using redundant streams and used as inputs*
*for single point simulations with the Community Terrestrial Systems Model (CTSM). Additional NEON observations are*
*used as input data to the model, including surface characteristics of vegetation (e.g., mapping to simulated plant*
*functional types, PFTs) and the soil properties (soil texture, organic matter content, and depth to bedrock, if < 2m).*
*Simulations with CTSM are conducted in CESM-Lab, a computing environment that runs in a container or with cloud*
*computing resources, which includes model code and analysis tools. Simulated data is compared with observed fluxes*
*using visualization scripts that are provided within CESM-Lab to improve both observed data products, model*
*parameterization, and model processes representation.*



## 2. Methods

### 2.1 NEON Data

NEON is a research network comprising 81 monitoring sites (47 terrestrial, 34 aquatic) that are collecting standardized, open data across the major ecosystems of the United States (Table S1). NEON's data products are highly complementary to land models, providing high quality and standardized data for soil, vegetation, and atmosphere states and fluxes across vast spatiotemporal scales with high throughput instrumented systems data and spatially expansive remote sensing data (Hinckley et al. 2016; Balch et al. 2020; Durden et al. 2020). Each of the 47 NEON terrestrial sites includes an EC tower to determine the surface-atmosphere exchange of momentum, heat, water, and $CO_2$, alongside meteorology (precipitation, wind speed, humidity, temperature), atmospheric composition (water vapor and $CO_2$ concentrations and isotopic ratios), and soil sensor assemblies (Metzger et al. 2019). In this preliminary effort to bring NEON measurements and NCAR modeling together we use NEON data for: 1) Meteorological inputs that are gap filled and provide local atmospheric boundary condition inputs to CTSM; 2) Surface characteristics of soil properties and vegetation; and 3) Eddy covariance fluxes to compare observed and simulated results (Fig. 1, Table 1), with prototype data available through the NEON data portal (NEON 2023).

*Table 1. NEON data product name, data product use in CTSM, NEON data product ID, and Digital Object Identifier (DOI). Data products were used for meteorological inputs and surface characterization, which are inputs needed to run CTSM, and for model evaluation.*

| Data Product Name | Data Product Use | Data Product ID | DOI |
|---|---|---|---|
| Precipitation | Meteorological input | DP1.00006.001 | https://doi.org/10.48443/6wkc-1p05 |
| Relative humidity | Meteorological input | DP1.00098.001 | https://doi.org/10.48443/w9nf-k476 |
| Shortwave and longwave radiation (net radiometer) | Meteorological input | DP1.00023.001 *DP1.00024.001 *DP1.00014.001 | https://doi.org/10.48443/stbf-bh38 https://doi.org/10.48443/8a01-0677 https://doi.org/10.48443/hv8e-5696 |
| Barometric pressure | Meteorological input | DP1.00004.001 *DP4.00200.001 | https://doi.org/10.48443/zr37-0238 https://doi.org/10.48443/7cqp-3j73 |
| Wind speed | Meteorological input | DP4.00200.001 *DP1.00001.001 | https://doi.org/10.48443/7cqp-3j73 https://doi.org/10.48443/77n6-eh42 |
| Air temperature | Meteorological input | DP4.00200.001 *DP1.00003.001 | https://doi.org/10.48443/7cqp-3j73 https://doi.org/10.48443/q16j-sn13 |
| Forcing height | Meteorological input | DP4.00200.001 | https://doi.org/10.48443/7cqp-3j73 |
| Soil physical and chemical properties, Megapit | Surface characterization | DP1.00096.001 | https://doi.org/10.48443/10dn-8031 |
| Dominant vegetation type | Surface characterization | Manually Assigned | |
| Bundled data products - eddy covariance | Model Evaluation | DP4.00200.001 *DP1.00023.001 | https://doi.org/10.48443/7cqp-3j73 |



| Net radiation | Model Evaluation | DP1.00023.001<br>*DP1.00014.001 | https://doi.org/10.48443/stbf-bh38<br>https://doi.org/10.48443/hv8e-5696 |
|---|---|---|---|
| Photosynthetically Active Radiation (PAR) | Model Evaluation | DP1.00024.001<br>*DP1.00023.001<br>*DP1.00014.001 | https://doi.org/10.48443/8a01-0677<br>https://doi.org/10.48443/stbf-bh38<br>https://doi.org/10.48443/hv8e-5696 |
| Direct and Diffuse Radiation | Model Evaluation | DP1.00014.001 | https://doi.org/10.48443/hv8e-5696 |
| Soil water content and water salinity | Model Evaluation | DP1.00094.001 | https://doi.org/10.48443/ghry-qw46 |

*Indicates the data product was used in the redundant stream gap-filling to fill primary data product
*2.1.1 Meteorological inputs*
Generating the gap-filled meteorological data that are required for single-point simulations with
land models can be time consuming and requires expertise in micro-meteorology that land model users
and developers may not have. Thus, the modeling community historically relied on external efforts like
FLUXNET synthesis databases to provide gap-fill meteorological measurements at eddy-flux sites (e.g.,
La Thuile or FLUXNET2015; Pastorello et al 2020). Downloading and processing these datasets into a
format that is usable by the model is also time consuming, and often the flux measurements are not
paired with information about local vegetation or soil properties that are easy to discover or digest.
Collectively, these factors create barriers for use and latencies in updating the EC observational data that
are used in single point simulations. The NCAR-NEON system aims to remove some of these barriers.
NEON meteorological input data used to run CTSM are summarized in Table 1, and gap-filled
using publicly available code (Table 2). While NEON is highly standardized, a few differences in
instrumentation exist between NEON Core (representative of the predominant natural ecosystem of each
respective Domain) and gradient sites (representing other endmember conditions in each respective
Domain). For example, core NEON sites measure precipitation with Double-fenced Intercomparison
Reference gauges, while gradient sites all have tipping buckets (Metzger et al. 2019). Accounting for
these site-specific sensor configurations and variation in their associated data streams is the first step in
providing usable meteorological inputs to CTSM. The meteorological inputs to CTSM must be continuous,
therefore, additional gap filling of missing data is required. Additionally, the EC system collects data
necessary to calculate fluxes of energy, water vapor, and $CO_2$. The NEON site design builds in some
redundancy in observations with profiles of incoming radiation, wind, temperature, water vapor, and $CO_2$
concentrations measured at different heights on each NEON tower (Metzger et al. 2019). These data
redundancies allow for a robust initial gap-filling using linear regressions among the primary and
redundant data streams to correct for instrument or location differences. For example, if wind speed or air
pressure measurements from the tower top are missing, we gap-fill with the value from the redundant
data stream (typically measured at a lower tower height) corrected by the linear relationship with the
primary sensor data. If multiple redundant data streams are available, the best fit regression with data
available is used to determine the gap-filled value for each missing data point.



After gap-filling using related data stream regression, some range thresholds and proper unit

conversions are applied to prepare the meteorological data for processing through the ReddyProc R
package following the gap-filling workflow outlined in Wutzler et al. (2018). After using related data stream
regressions, the meteorological data are checked for additional gaps, and gap-filling is performed using
one of three additional gap-filling methodologies that include look-up table (Falge et al. 2001), mean
diurnal course, and marginal distribution sampling (Moffat et al. 2007; Reichstein et al. 2005). The gap-
filling method is tracked and provided as a flag with the data to allow users to assess data with various
methodology restrictions. The meteorological data streams are then converted to units required by CTSM
and output to cloud storage in netCDF format with associated metadata to fully describe data provenance
and formatting. At most sites data coverage spans January 1, 2018, through December 31, 2021, but as
more NEON data are collected these files will also be updated in near-real time, thus removing barriers
associated with processing flux tower data and reducing latencies in using new data as they are
collected. Tables S1 and S2 provides a list of all the sites where input data have been successfully gap-
filled and notes any potential data quality issues.



***Table 2*** *List of helpful websites created for the NCAR-NEON system, their contents and a url address for each. All sites*
*were accessed Feb 13, 2023. *Note we intend to provide permanent urls for these sites in the final published*
*manuscript.*

| Name | contents | url |
|---|---|---|
| Project home page | Main landing page for users interested in learning more about the project | https://ncar.github.io/NEON-visualization/ |
| Tutorial | Tutorial that introduces running CTSM at NEON tower sites in the CESM-Lab container. | https://ncar.github.io/ncar-neon-books/notebooks/NEON_Simulation_Tutorial.html |
| Interactive visualizations | Interactive plots that allow users to explore data produced by the NCAR-NEON system without running the model or downloading data. | https://neon.herokuapp.com/neon_dashboard |
| Processing NEON data | Docker image with scripts used for gap filling meteorological data, flux partitioning, and formatting NEON datasets. | https://quay.io/repository/ddurden/ncar-neon |
| DiscussCESM Forum | Discussion forum bulletin boards for questions related to CESM including CESM-Lab and CTSM. | https://bb.cgd.ucar.edu/cesm/ |
| CTSM repository | Code base, technical documentation and information related to CTSM | https://github.com/ESCOMP/CTSM |
| NEON Prototype Data | NEON prototype datasets, which include the gap filled meteorological data for flux partitioned data used for model input and evaluations | https://data.neonscience.org/prototype-datasets/0a56e076-401e-2e0b-97d2-f986e9264a30 |





*2.1.2. Surface characteristics of soil properties and vegetation*
Basic information on edaphic properties is needed in the pedotransfer functions that describe soil
thermal and hydraulic properties in CTSM. Although NEON has several soil sampling datasets, we used
information from the Megapit characterization of soil physical and chemical properties in CTSM because it
contains more information about deep soil horizons (> 1 m depth; Table 1) from a single soil pit at each
site. Megapit samples were collected by pedogenic soil horizon down to 2 m or restrictive feature and
analyzed for several properties including total soil carbon concentration, calcium carbonate concentration,
bulk density, coarse fragments, soil pH, and texture. Soil organic carbon stocks used in CTSM were
estimated for each soil horizon by calculating organic carbon concentrations (after subtracting carbonates
from total carbon measurements) and multiplying by bulk density.
Currently, the CTSM simulations are run with a single plant functional type (PFT) at each NEON
site (Table S1). We acknowledge that this belies the diversity in vegetation that is present at NEON sites,
but it provides a tractable starting point for further investigation into developing more sophisticated site- to
regional-scale parameterizations and representations of biotic diversity with CTSM. The dominant PFT at
each NEON site was assigned at the location of each EC tower using expert assessment that was
informed by NEON vegetation surveys. Information on soil properties and dominant vegetation types are
output as .csv files to public-access cloud storage buckets for use by CTSM (Figs. 1; Sect. 2.3).
*2.1.3 Independent model evaluation*
The EC flux data (energy, water vapor, and $CO_2$) are time regularized and quality assurance and
control (QA/QC) are applied. The QA/QC applied includes removing data when quality flags are raised,
removing $CO_2$ data when the field calibration algorithm cannot be applied, applying range thresholds, and
applying a despiking routine to remove outliers (Brock, 1986; Starkenburg et al. 2016. The data are gap-
filled using the ReddyProc methodology outlined in Sect. 2.1.1. The vapor pressure deficit (VPD) is
derived from the difference between actual and saturated vapor pressure, while gross primary production
(GPP) is calculated from net ecosystem exchange (NEE) using the nighttime flux partitioning method of
Reichstein et al. (2005). The data, quality flags, and metadata are formatted and provided at 30-minute
intervals as netCDF files for comparison with modeled fluxes. Finally, NEON continuous soil moisture
data were compared with model simulations for two sites. Since the soil moisture sensors were
reconfigured with different calibration coefficients during the 2018-2021 validation period, which
introduced step changes in NEON's soil moisture data product (Table 1), the raw sensor measurements
were back-calculated and consistent soil-specific calibration coefficients were subsequently applied over
the entire measurement period (Ayres et al. 2021) prior to comparison with CTSM data. Only values that
passed quality tests were used. In future work we aim to provide standardized soil moisture data for more
sites across the Observatory.



**2.2. NCAR modeling**

Numerical models of weather and climate have long been recognized as essential research tools
to advance atmospheric science. Land surface fluxes of energy, moisture, and momentum, required to
solve the equations of atmospheric physics and dynamics, are controlled by heat and water storage in
soil, as well as the physiology of plants and their organization into canopies of leaves. Consequently,
models of soil-plant-atmosphere processes are required to provide the necessary surface fluxes. Indeed,
the first numerical weather prediction model included mathematical equations for soil temperature, soil
moisture, the stomata on leaves, and envisioned canopies as a film of leaves covering the surface
(Richardson 1922). As science progressed from models of atmospheric general circulation to climate
models and now, Earth system models, the role of terrestrial ecosystems in climate processes has come
to the forefront. The terrestrial components of ESMs, such as CTSM, have improved ecological processes
representation and now include biogeochemical cycles, wildfires, and land use and land cover change
(Bonan 2015, 2019; Lawrence et al. 2019). This evolution in the Earth system sciences is evident in 40+
years of scientific research linking weather, climate, and land modeling at NCAR, from pioneering initial
model implementations (Deardorff 1978; Dickinson et al. 1986, 1993; Bonan 1996) to community-based
model development (Oleson et al. 2004, 2010, 2013; Levis et al. 2004; Lawrence et al. 2019) that
continues to engage ecological and environmental sciences communities in CTSM development and
application. As more ecology and biogeochemistry are added to the models (Fisher and Koven, 2020),
the notion of climate prediction is expanding to Earth system prediction, including terrestrial ecosystems
and biotic resources (Bonan and Doney 2018). These models have also become important tools for
scientific discovery by identifying the ecological processes that affect climate (e.g., photosynthetic
temperature acclimation; Lombardozzi et al. 2015) and to advance theory at the macroscale (e.g.,
developing a theory of ecoclimatic teleconnections; Swann et al. 2018). With the new NCAR-NEON
system tools described here, we aim to expand engagement and accessibility with the ecological and
environmental sciences communities to continue testing, evaluating, and improving terrestrial process
representation within CTSM. This will improve our understand of how ecosystems function within the
Earth system, including the regulation of carbon, water, and energy fluxes that affect climate.
*2.2.1 Containerized version of CESM-Lab*

CESM has a long history of being freely and openly available to users (Hurrell et al. 2013;
Danabasoglu et al. 2020), yet several barriers related to training, cyberinfrastructure, and data integration
have hampered its adoption by a wide range of researchers. Even with open-source software, porting
CESM to a new computer also requires the new computing system can compile model source code and
has all the necessary input data and library dependencies. To address these computing challenges,
NCAR recently developed CESM-Lab, which is a pre-configured and standardized environment that
contains CESM and Jupyter-Lab. CESM-Lab is available via a Docker container and distributed via
DockerHub (Table 2). The containerized version of CESM-Lab, and containers in general, give



researchers the capability to package and distribute source code, libraries, dependencies, and system
settings as one unit – thereby ensuring reproducibility. Using the containerized system, CESM-Lab can
be used on any computing system, even a laptop or a cloud platform, to allow researchers to easily run
CESM and its component models. The NCAR-NEON system uses CESM-Lab capabilities to run single
point CTSM simulations at NEON sites.
*2.2.2 Single point CTSM simulations*
The workflow for running single-point CTSM simulations requires several steps that can be error-
prone and time-consuming, particularly when using EC tower or other site-level data to drive simulations.
To facilitate using NEON data in CTSM simulations we made several modifications to simplify this
workflow. When users create a new simulation, the system queries NEON public-access cloud storage
buckets and downloads available data into a designated directory (Sect. 2.3). For each NEON site, this
includes a surface dataset that reflects soil properties and the dominant vegetation (Table 1),
meteorological data used to drive the atmospheric conditions, and an initial conditions file with
equilibrated carbon, water, energy, and nitrogen states and fluxes. Initial conditions at each NEON site
were generated by cycling over the meteorological data at each site for 200 years in accelerated
decomposition (AD) mode and another 100 years in normal, or post-AD mode, or until biogeochemical
states reached steady state (when ecosystem C pools change by < 1g C $m^{-2}$ $y^{-1}$; this is standard protocol
for equilibrating the model state, Lawrence et al. 2019). Colder sites, especially those in Alaska, took
longer to reach these steady state conditions.
The NCAR-NEON system uses a top-level Python code called 'run_neon' that simplifies
downloading the preconfigured datasets and automatically creates, builds, and runs cases for individual
and multiple NEON sites. The Python script, which also resides in the CTSM repository (Table 2),
includes several command-line arguments and options for automatically running spin-up and transient
simulations. Collectively, these features dramatically improve CTSM site simulation accessibility, facilitate
the use of new NEON data, reduce potential errors in configuring the CTSM case at NEON tower sites,
and enable users to run simulations at multiple NEON sites. While users of the system can now easily
generate their own data, NCAR provides model simulation data at each of the tower sites that are
available on the NEON public-access cloud storage bucket (Sect. 2.3). Simulation data are generated at
a 30-minute time step and are aggregated into daily netCDF files.
*2.2.3 Tutorials, analysis, and visualization*
Three interactive tutorials are available to guide users through the new NCAR-NEON system
(Table 2). The first tutorial helps system users to access CESM-Lab using Docker, which will ultimately
allow the user to run CTSM simulations at NEON sites on their local computing system. The first step
requires that users download Docker from the company website. This step is potentially challenging, as
Docker is an externally controlled application and some recent Docker updates do not work with older





computing systems. We provide links to additional resources to help the user navigate these potential
problems and offer a resource for asking questions about containers through the CESM discussion forum
(Table 2). After downloading and installing Docker, users are guided through downloading, running, and
connecting to the CESM-Lab container and accessing the NEON tower simulation and visualization
tutorials.

The second tutorial is a Jupyter Notebook that guides users through running CTSM simulations

for NEON flux tower sites. The beginning of this tutorial provides a short description about CTSM and its
component models, as well as resources for finding additional information. The process of running a
simulation at NEON tower sites has been streamlined into the 'run_neon' script (see Sect. 2.2.2) that can
be called with a single line of code after the user defines a NEON tower site. The simulation itself
downloads approximately 2.5 GB of input data and takes several minutes or more to complete, depending
on the speed of the internet connection and computing system being used. After the simulation
completes, the user is pointed to where the model data are stored and has the option to generate plots of
soil temperature and moisture profiles for one year of the simulation.

The third tutorial guides users through analyzing and evaluating model simulations against

observed NEON flux tower measurements. This tutorial requires a successfully completed NEON tower
simulation from the previous simulation tutorial. The user selects their site and the year of interest and is
guided through loading and opening the model data files, as well as downloading EC data for evaluation
from the NEON server and loading and opening the files. Next, the tutorial guides users through
formatting, processing, and plotting simulation and flux tower data. Users generate plots of mean annual
and diel cycles of latent heat flux. Additional plots illustrate how CTSM partitions latent heat flux into
ground evaporation, canopy evaporation, and transpiration, as component fluxes are not available from
the observed data. Scatter plots are also created using simulated fluxes to illustrate the relationship
between component evaporation and transpiration fluxes and total latent heat flux on seasonal and
annual timescales. The tutorial explains the python tools used to process and plot the data and asks
probing questions about the results that tutorial users are exploring to help guide the user in thinking
about patterns in the data and consider how to compare model and flux tower data. Users are
encouraged to use the code available in this tutorial to explore other sites, years, and variables.
**2.3 Cyberinfrastructure to Facilitate Data Exchange and Interactive Visualizations**

Cyberinfrastructure for scientific data provides data handling and management functionality

including data storage, processing, transfer, security, and access. Cyberinfrastructure components
developed for the NCAR-NEON system include access-managed cloud storage for project data,
standards-based metadata generation enabling dataset search and discovery, and data exploration tools
for the user community. Datasets for the NCAR-NEON system are hosted in cloud object storage
providing secure web-enabled access to the data files (Fig. 2). Data files are grouped in the cloud storage
system into logical storage containers called buckets. Buckets that are granted public access allow




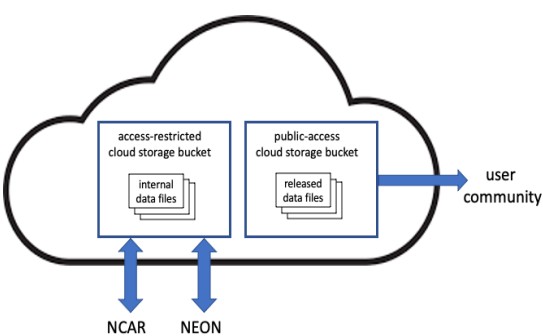

anyone on the Internet to download the data stored in them. Buckets protected with authentication mechanisms require users to have either individual account permissions on the bucket or an access key for the bucket and are meant for internal dataset sharing or staging data prior to public release.

Data exchange between NCAR and NEON within this system enables automated generation of datasets as well as collation of NCAR model outputs and NEON data. The initial data collation for NEON data products uses a container that sources all atmospheric forcing and model evaluation data from the NEON API, performs gap-filling, and formats the data for model ingestion with standardized metadata (Sect. 2.1). Simulation datasets from NCAR (Sect. 2.2) are

*Figure 2. A schematic representation of the cloud-based data management for the NCAR-NEON system. Internal data may include preliminary results, data shared for review within the project, or data staged for release. Released data files are available for public access to the user community and anyone on the Internet and include NEON meteorological inputs, NEON surface characterization data, CTSM surface datasets and initial condition files, NEON measurements used for model evaluation, and data from CTSM simulations that are used for interactive visualizations. Access-restricted cloud buckets require authentication to access files stored in them. Public-access cloud storage buckets provide open access to the files stored in them.*

automatically synced to NEON object storage in the cloud at scheduled intervals (Fig. 2). To facilitate
automated transfer of datasets between NCAR and NEON, a staging bucket is configured that allows file
uploads from authenticated users. An automated process moves files from the staging bucket to the
publicly available target bucket at scheduled intervals. Metadata describing scientific datasets using
standard vocabularies and formatting can be used by Internet search engines to facilitate dataset
discovery. JavaScript Object Notation for Linked Data (JSON-LD; https://www.w3.org/TR/json-ld) is a
human- and machine-readable open metadata standard. Schema.org defines a vocabulary of standard
HTML tags compatible with JSON-LD markup (Shepherd et al. 2022). A metadata generation component
for NCAR-NEON datasets is implemented in Python and uses the Binary Array Linked Data library
(binary-array-ld 2016) to generate JSON-LD metadata for NCAR-NEON netCDF files with the
Schema.org vocabulary.
Beyond these automated data exchanges, we also developed a Python-based interactive
visualization dashboard (Table 2) as a Graphical User Interface (GUI) that enables users to explore and
interact with model outputs and observations on-the-fly. This tool allows users to generate graphs and
statistical summaries comparing CTSM simulations and observational data for NEON sites without
downloading the observational data or running the model. This dashboard was developed using a
scientific Python stack, including Xarray, Bokeh, and Holoviews, which allows a developer to create a
user interface with widgets and visualization components inside a Jupyter Notebook. Users access a GUI
to select individual NEON sites, variables, and output frequencies to visualize. The tool offers different



types of interactive visualizations and statistical summaries based on users' selections. This interactive
visualization dashboard does not require specialist knowledge to operate; therefore, it can be used for
educational outreach activities and in classrooms. Moreover, users can interact with the dashboard using
a browser, so it is possible to interact with the plots via tablet or smartphone.
Data I/O and manipulation, particularly at the 30-minute frequency available in the NCAR-NEON
system, are typically computationally resource-intensive aspects of data access. I/O and calculations can
both benefit from parallel computing, which can process multiple subsets of a dataset simultaneously and
thereby enable efficient dataset access and operations. The back end for the visualization dashboard
uses dataset chunking for efficient access to netCDF file content. The Zarr format and library enable
generation of metadata providing chunked access to netCDF files (Miles et al. 2022). Zarr metadata for
daily files is combined into monthly files, reducing the number of files accessed for time intervals
spanning multiple days and thereby improving access efficiency. The Python Xarray library, which is used
to read the datasets, integrates with the Python Dask library for parallel computing and thus enables
loading and processing netCDF data chunks in parallel as Dask arrays. The Dask components that
Xarray uses use a local thread pool by default, and local threads incur minimal task overhead associated
with the parallel processing. Operations on the Dask arrays use the Python NumPy library for array
operations, and the NumPy implementation takes advantage of thread pool parallelism, enabling
efficiency improvements in dataset operations even on small (~100-200 KB) files.
**3. Results**
We illustrate features of the NCAR-NEON system with comparisons of observed and simulated
fluxes across diverse ecosystems that the Observatory spans. A subset of the sites highlighted in our
analysis are described in Table 3. The comparisons are intended to summarize the status of the project,
illustrate the data produced through this project, and highlight potential insights the data affords. We
recognize that there are rich opportunities to expand on these analyses, integrate additional
measurements, and improve modeled parameterization and representations of specific sites and
processes. Indeed, such contributions are encouraged from the community.



**Table 3** *Summary of site name, location, mean annual temperature (MAT), mean annual precipitation (MAP), and gross primary production (GPP) at a subset of NEON sites. Due to gaps in the observational estimates, mean annual GPP is for the full time series simulated by CTSM at each NEON site. All results are for 2018-2021 unless noted otherwise. The full list of results is shown in Tables S1, S2.*

| NEON Site ID | Site Name | Lat | Lon | MAT (°C) | MAP (mm y$^{-1}$) | GPP (gC m$^{-2}$y$^{-1}$) |
|---|---|---|---|---|---|---|
| BART | Bartlett Experimental Forest | 44.06516 | –71.28834 | 7.7 | 1213 | 1127 |
| HARV | Harvard Forest | 42.53562 | –72.17562 | 8.5 | 1405 | 1153 |
| STEI | Steigerwaldt-Chequamegon | 45.5076 | –89.5888 | 5.7 | 660 | 1109 |
| KONZ | Konza Prairie Biological Station | 39.1007 | –96.56227 | 12.9 | 617 | 1158 |
| SRER | Santa Rita Experimental Range | 31.91068 | –110.83549 | 20.4 | 329 | 360 |
| ABBY | Abby Road | 45.762378 | –122.329672 | 10.1 | 2043 | 1906 |

Annual climatologies of site level data provide comparisons of measured and simulated fluxes. Site level simulations with CTSM received inputs of incoming shortwave and longwave radiation measured at NEON EC towers (Table 1), but the model calculates reflected shortwave radiation and outgoing longwave radiation based on albedo and surface temperature. Accordingly, net radiation is a useful metric by which to compare observed and simulated fluxes. Since net radiation is a driver of numerous ecosystem fluxes, identifying biases can help to explain biases in other fluxes. We look at a climatology of daily mean net radiation that is simulated over the NEON record. Results shown here for Bartlett Experimental Forest (BART; Fig. 3a) suggest that the model adequately captures the seasonal cycle of net radiation at this temperate deciduous forest site. (Fig. S1 shows a similar climatology for a boreal forest site at DEJU).



**Figure 3** *Climatology of daily mean NEON measurements (orange) and CTSM simulations (blue) at the Bartlett Experimental Forest in New Hampshire (BART). Points show the daily mean (a) net radiation; (b) sensible heat flux; (c) latent heat flux; (d) gross primary production (GPP); and (e) net ecosystem exchange (NEE). Shading shows the standard deviation of daily average data for 2018-2021.*

Users can also compare latent and sensible heat fluxes that are simulated by the model and observed at EC towers. At BART we see that CTSM tends to underestimate sensible heat fluxes, while

2000

2000

2000

2000

2000

2000

2000

2000



overestimating latent heat fluxes, especially during the summer months (Fig. 3b-c). Such biases in the
evaporative fraction (the ratio of latent heat flux to the sum of latent and sensible heat fluxes) of turbulent
fluxes are common in land models, including CTSM (Best et al. 2015; Wieder et al. 2017) and the NCAR-
NEON system. The inconsistencies at BART could reflect model biases in stomatal conductance or leaf
area index (LAI) and deserves further investigation. Future work can leverage data from PhenoCam data
(Richardson et al. 2018) and stable isotope measurements at NEON towers (Finkenbiner et al. 2022;
Moon et al. 2022) to better understand LAI and stomatal conductance, respectively.
Comparing measured and simulated carbon fluxes provides insights into model parameterizations
and can be used to estimate missing observational data. Carbon fluxes from CTSM simulations can be
compared to data from NEON EC towers: Net ecosystem exchange (NEE) data are measured at the
NEON EC towers while GPP is a modeled product that is derived from statistical relationships, here using
the nighttime flux partitioning method of Reichstein et al. (2005). By contrast, models like CTSM first
simulate GPP based on leaf level photosynthetic rates that are scaled to the canopy with simulated LAI.
Subsequently, NEE is calculated after subtracting ecosystem respiration fluxes from GPP. Results at
BART suggest that CTSM generally captures the timing and magnitude of GPP fluxes at the site (Fig. 3d);
although attention to phenology, especially environmental controls and interannual variability of leaf out
and senescence are likely warranted (Birch et al. 2021; Li et al. 2022). The climatology of NEE fluxes
simulated by CTSM shows biases during the spring and autumn when the model simulated a land source
of $CO_2$ to the atmosphere (Fig. 3e) due to high ecosystem respiration fluxes. Moreover, the land sink of
$CO_2$ in the summer appears to be weaker in CTSM simulations than the NEON observations at the BART
tower (Fig. 3e). Since the magnitude of GPP is similar in the model and observations, the underestimated
summer NEE is possibly due related to high biases in simulated ecosystem respiration fluxes. Diagnosing
the source of this model biases is challenging, in part due to the interconnectivity of simulated processes
and the limited capacity to measure such processes. Deeper insights may be afforded by taking a closer
look at results with higher temporal frequencies.
NEON tower data are simulated in near-real time within the NCAR-NEON system, with data
available to simulate most towers starting in 2018 through the most recent full year, here 2021. Figure 4
shows daily mean carbon fluxes, NEE, that are measured and simulated for the Konza Prairie Biological
Station (KONZ), where the NEON tower is in an unplowed tallgrass prairie in Kansas, and Steigerwaldt
Land Services (STEI) site, where the NEON tower is located in an early successional aspen stand in
Wisconsin. Positive NEE fluxes show net carbon release from land to the atmosphere, while negative
fluxes indicate carbon gain into ecosystems. Looking at the full data record shows several notable
features of NEON measurements and CTSM simulations. Data gaps in NEON measurements are most
common during the early operation of the observatory (Aug-Oct of 2018 at STEI) and in the early months
of the COVID-19 pandemic, when field crews could not travel to field sites to maintain equipment (Apr–
June of 2020 at STEI). Across the observatory the NEON EC measurements have greater than 70% data
coverage, up from less than 40% data coverage at the start of observatory operations. The current NEON



EC data coverage aligns with that of the FLUXNET2015 dataset (van der Horst 2019). Second, although
EC is directly measuring NEE, mean daily NEON observations show high variability at both sites. Finally,
NEON EC towers measure both storage and turbulent fluxes, but results shown here omit the storage
component. Storage fluxes contribute to uncertainty in measured NEE fluxes, which may (or not) be large
for individual sites at different times of year.

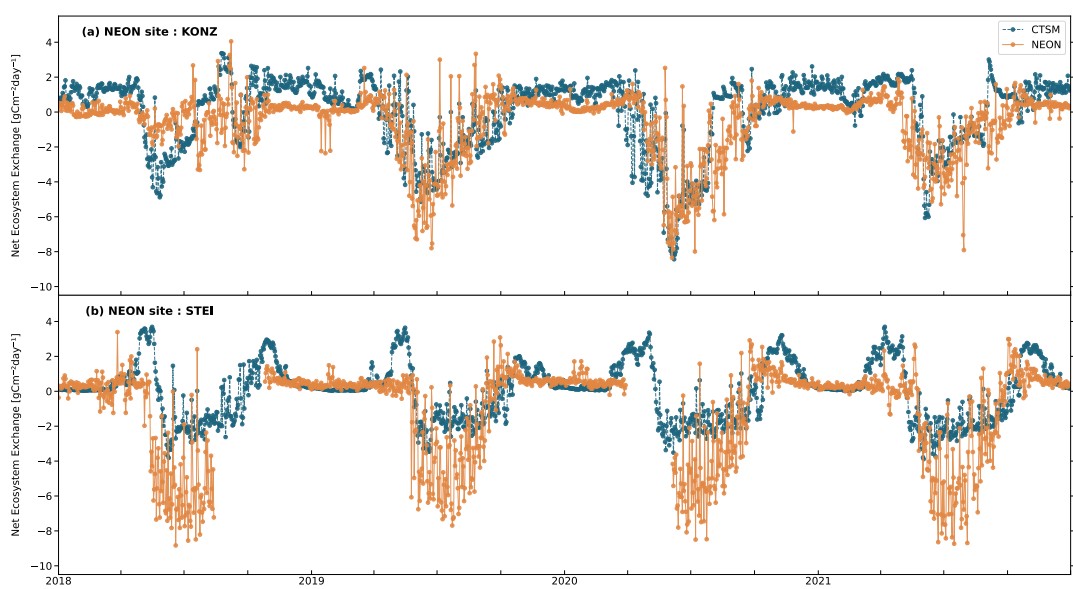

*Figure 4 Full time series of daily mean net ecosystem exchange (NEE) from NEON measurements (orange) and CTSM*
*simulations (blue) at the (a) Konza Prairie Biological Station in Kansas (KONZ) and (b) Steigerwaldt Land Services site*
*in Wisconsin (STEI). Positive NEE fluxes show net carbon release from land to the atmosphere, while negative fluxes*
*indicate carbon gain into ecosystems.*

The NEE fluxes that are simulated by CTSM are calculated as the differences in GPP and
ecosystem respiration fluxes, which includes both autotrophic and heterotrophic respiration. These
component fluxes are much larger, depend on simulated ecosystem states (LAI, vegetation biomass, and
soil organic carbon stocks) and have associated environmental sensitivities (e.g., temperature,
precipitation, etc.). Thus, biases in these component fluxes can potentially transmit biases to simulated
NEE fluxes (Figs. 3-4). For example, CTSM simulations show periods of positive NEE during the spring
and fall that are not evident in NEON observations. The seasonal biases in NEE could result from an
underestimation of GPP during the shoulder season caused by phenological mismatches in simulated
and observed LAI, or result from only simulating a single plant functional type in CTSM. Alternatively,
NEE biases could result from higher than observed soil respiration rates in the model that reflect potential
biases in total soil C stocks or the temperature sensitivity of heterotrophic respiration. Finally, the CTSM
simulations were equilibrated to steady state conditions, meaning that annual NEE averaged over the
simulation period will be zero. The real ecosystems being measured at NEON sites, however, have



historical legacies – KONZ is burned periodically and STEI is an aggrading forest site – and do not
necessarily meet these same steady state assumptions. Collectively, this points to rich opportunities to
learn about the ecosystems being measured by NEON observations and the processes that are important
to represent in models like CTSM.
We calculated summary statistics of CTSM simulated bias (Fig. 5) and root mean square error
(RMSE; Fig. S2) in ecosystem fluxes, compared to NEON observations. Biases in GPP and NEE are
relatively low in the Great Plains and Intermountain West but are larger in the Eastern US. Specifically,
NEE is biased high east of the Mississippi, while GPP biases are largest in the Southeastern US. CTSM
typically has high biases in sensible heat fluxes and concurrent low biases in latent heat flux. Some sites,
particularly grasslands (e.g., CPER, OAES, and SJER), do not follow this general pattern. We therefore
probed precipitation data from NEON, which appear to have significant biases at some grassland sites
(discussed in Sect. 4.1) and contribute to artificially high biases in CTSM simulations at these sites.

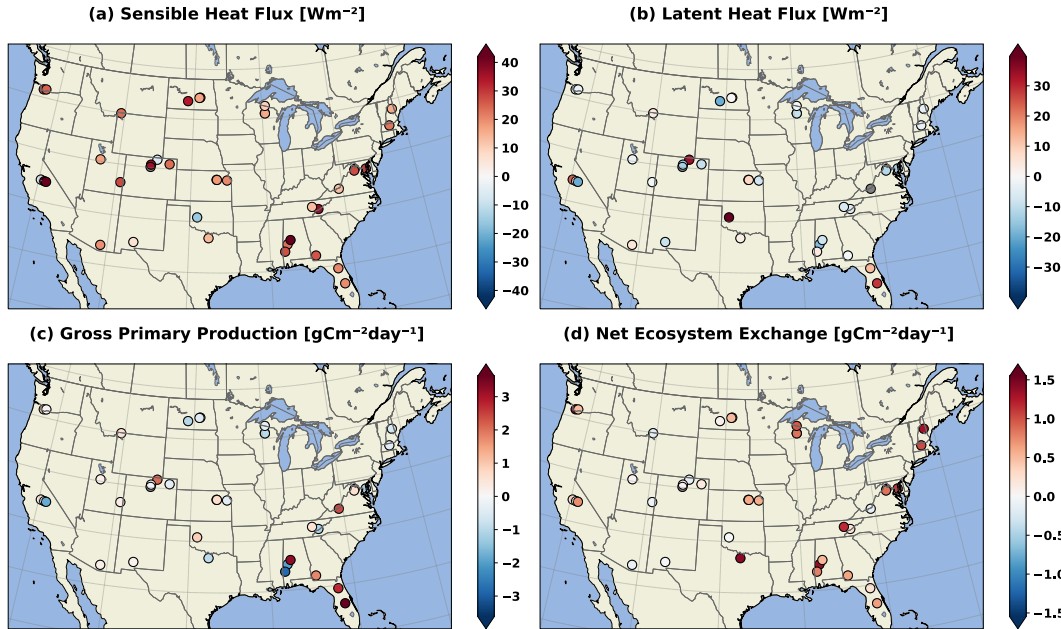

**Figure 5** *Maps showing location of NEON site in the conterminous United States and annual biases in fluxes that are*
*simulated by CTSM for: (a) sensible heat flux (W m⁻²); (b) latent heat flux (W m⁻²); (c) gross primary production (GPP,*
*gC m⁻² day⁻¹); and net ecosystem exchange (NEE, gC m⁻² day⁻¹) over the observational record (2018-2021), unless*
*otherwise noted in Table S2.*

Additional insights into potential sources of biases in data-model comparisons can be provided by
looking deeper into component fluxes of latent heat at higher temporal frequencies. The NEON EC towers
provide 30-minute measurements of total latent heat fluxes, but latent heat fluxes in CTSM can be
partitioned into contributions from canopy transpiration, canopy evaporation, and soil evaporation. For

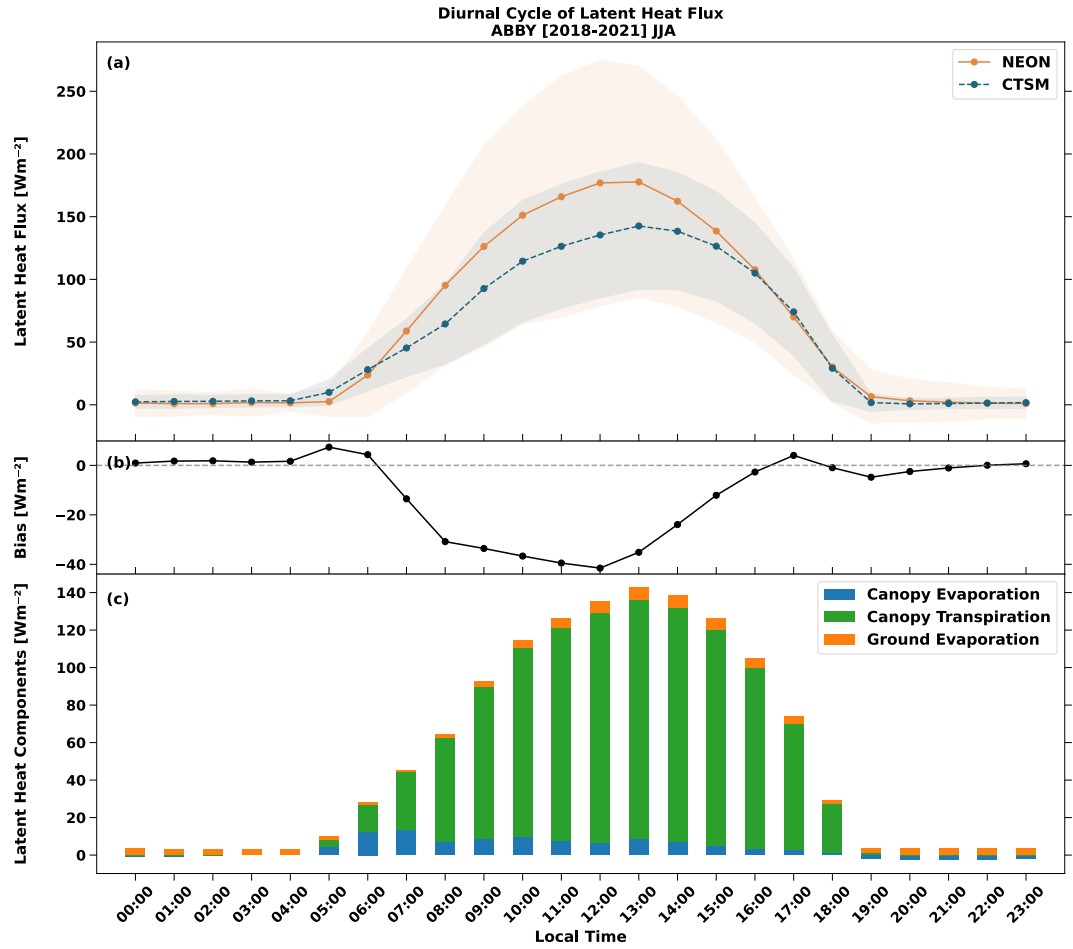

**Figure 6** *Diel cycle of summertime (June, July, and August, or JJA) latent heat flux at the Abby Road site in Washington (ABBY). Panels show: (a) mean half hourly fluxes (2018-2021 mean ± 1σ) for NEON measurements and CTSM simulations (orange and blue lines, respectively); (b) CTSM model bias relative to the observations (W m⁻²); and (c) partitioning of latent heat into fluxes that are simulated by CTSM, which includes canopy evaporation, canopy transpiration, and ground evaporation (blue, green, and orange bars, respectively). Additional visualizations showing all sites and seasons are available on the interactive visualizations web site (Table 2).*

example, the CTSM simulations show temporal biases in both the timing and magnitude of mean diel cycle of summertime (June, July, and August, or JJA) latent heat fluxes at the NEON Abby Road site (ABBY; Fig. 6). The bulk of daytime latent heat fluxes simulated by the model are coming from canopy transpiration fluxes, suggesting that the representation of stomatal conductance does not respond correctly to atmospheric conditions or plant water availability. We also note that this site experienced two very strong heatwaves in the summers of 2020 and 2021. Additional measurements of soil moisture, LAI, or sap flux could help test, evaluate, and improve various model parameter values and parameterizations to produce results that are most consistent with observed fluxes.



Light response curves (Fig. 7) illustrate how canopy photosynthesis responds to changes in the

radiation environment. At forested sites, CTSM tends to overestimate GPP at low light levels,
underestimate GPP under full irradiance and simulate lower variance in GPP across a range of high
incident radiation; this pattern is illustrated in Fig. 7a for Harvard Forest. At the Santa Rita grassland site,
GPP is biased high in most irradiance bins, although is comparable to observed estimates of GPP at full
irradiance (Fig. 7b). As GPP is the driver for carbon fluxes and plant-mediated water fluxes in CTSM,
inaccurate responses to light environment affects several processes, including NEE and transpiration,
which is a primary driver of mid-day (Fig. 6c) and summertime latent heat flux.

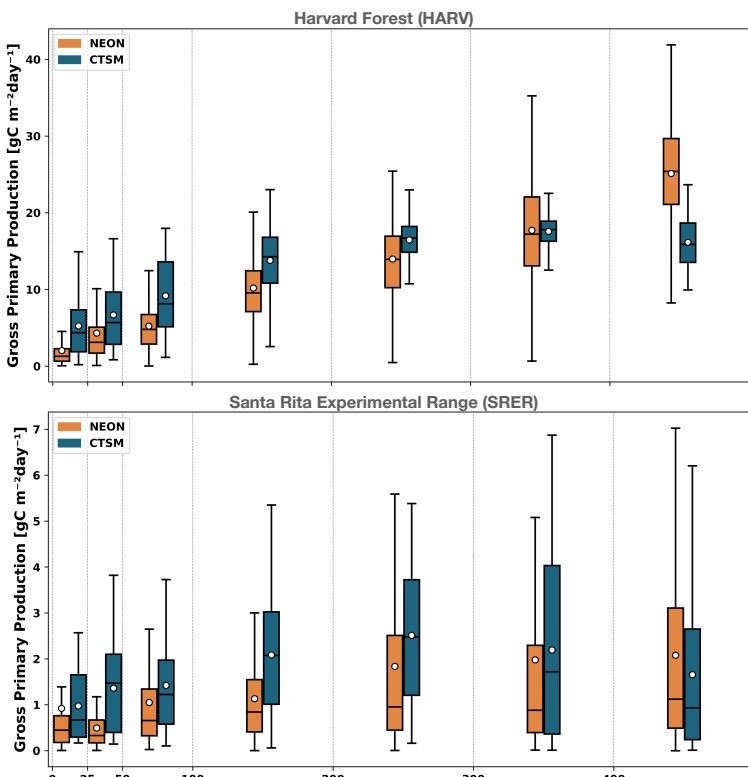


*Figure 7* *Box-whisker plots showing light response curves, the relationship between gross primary production (GPP)*
*and incident shortwave radiation, that are derived from NEON measurements and CTSM simulations (orange and blue,*
*respectively) at (a) Harvard Forest (HARV) and the (b) Santa Rita Experimental Range (SRER). Data represent 30-*
*minute measurements that are binned by incident shortwave radiation levels observed at NEON sites over the*
*observational record in July (2018-2021). Boxes show the mean (dots), median (line), interquartile range (boxes). The*
*whiskers extend from the boxes (showing first and third quartiles) by 1.5 times the interquartile range (Q3-Q1). Note*
*differences in the scale of the y-axis.*

Finally, there are opportunities to use data from CTSM simulations to augment NEON

measurements. For example, measurements of soil moisture are important for calculating soil $CO_2$ fluxes





from NEON sites, but the soil moisture probes currently deployed at NEON sites do not always provide
reliable measurements. For example, at the Abby Road site soil moisture observations have phases of
erratic measurements, are missing at depth throughout much of 2020 and 2021, and have large offsets
when instruments were calibrated (Fig. 8, Fig. S3). By contrast, CTSM provides continuous datasets that
could be used to gap fill or augment ongoing NEON soil moisture measurements, although simulated
data may need to be bias corrected. Similarly, soil moisture controls aspects of plant phenology in CLM,
meaning that soil moisture measurements could help constrain or explain potential biases in simulated
LAI and ecosystem fluxes. At ABBY, both CTSM simulations and NEON observations show similar
temporal patterns – a dry-down of soil moisture during the dry summer months and followed by wetter fall
winter and spring months (Fig. 8; Fig. S3), although CTSM simulates wetter soils than observed at the
NEON site.

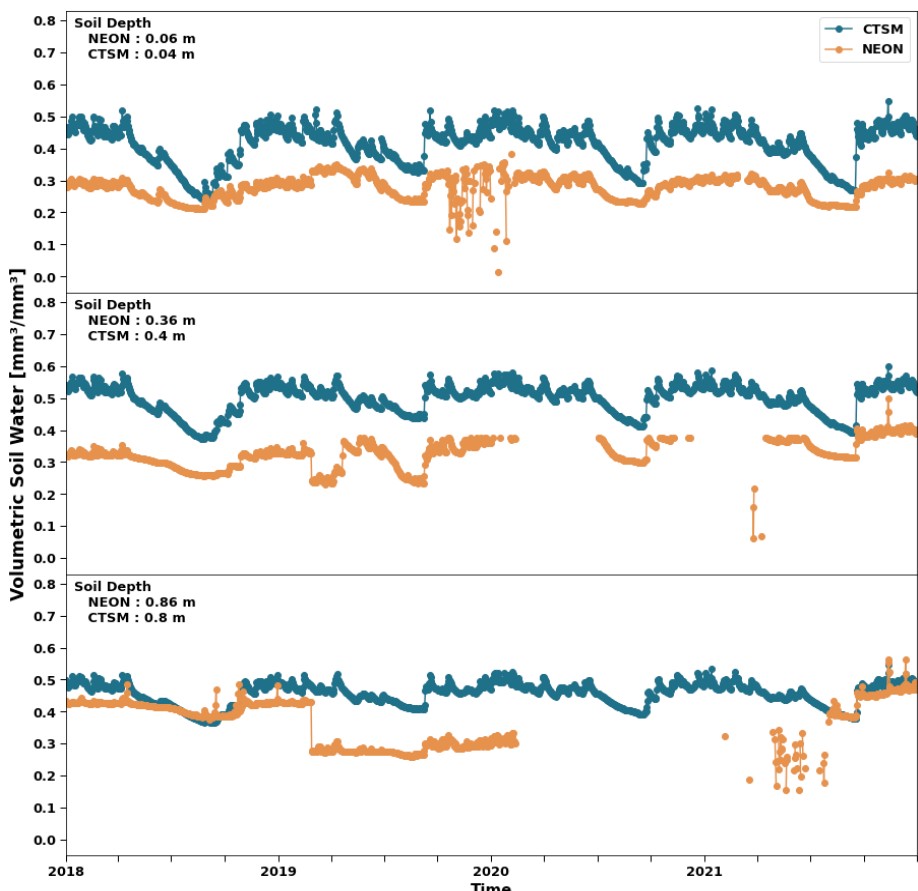

**Figure 8** *Time series of volumetric soil moisture profiles that are simulated by CTSM simulations (blue) and*
*measured by NEON (orange) at different depths in soil plot 3 at the Abby Road site in Washington (ABBY) from*
*2018-2021.*



## 4. Discussion

The NCAR-NEON system links models and measurements to provide a powerful suite of tools to
understand ecosystem properties and processes through space and time. In addition to facilitating the
integration of measurements and modeling, a major focus of this work is to enable new opportunities for
research and education by expanding access to and interaction with NCAR models and NEON data. The
user community can access quality-controlled and gap-filled NEON meteorological and EC flux data as
prototype datasets through the public-access cloud storage buckets that supports the NCAR-NEON
system or the Prototype Data section of the NEON Data Portal (Table 2). Additionally, the NCAR-NEON
system streamlines running NCAR's CTSM model and simplifies access through the containerized
CESM-Lab platform, bypassing the logistical challenges of porting CTSM to different computing systems.
It also creates customized model input data that include local site characterizations of soil and vegetation
using NEON data products. These capabilities allow researchers to focus their time on customizing CTSM
and integrating additional NEON datasets to address research questions. Combined with the visualization
software provided in the tutorials, the NCAR-NEON system also facilitates opportunities for teaching
about land-atmosphere interactions, ecology, and land modeling. Below we discuss some of the
synergistic enhancements this collaboration makes for NEON measurements and NCAR models as well
as opportunities that the NCAR-NEON system enables for research and teaching.

**4.1 Synergistic enhancements of NEON measurements and NCAR models**

The NCAR-NEON system is a collaborative partnership between observationalists and modelers
that enhances both NEON's measurements and NCAR's models. One typically thinks of observations as
improving models, but the reverse can also happen in which models inform and augment the collection of
measurements. For example, models require continuous meteorological input data, so gap filling the
missing meteorological data required to run CTSM was paramount to the success of the project. A new
prototype data product provided by the project is a continuous time series of meteorological data at each
NEON location. Comparison of modeled and measured EC fluxes identified QA/QC improvements to the
meteorological data needed for the model simulations, and similarly improvements to the processing of
the raw EC fluxes to compare with model results.
One issue raised in the simulations is the estimation of precipitation at grassland sites. NEON has
experienced issues where small amounts of noise in the raw data cause spurious trace precipitation to be
recorded at all primary precipitation sensors. Because secondary and throughfall precipitation buckets are
unaffected, there is a redundant data stream at forested sites, but these are unavailable for grassland
sites. An updated algorithm is expected to resolve the spurious trace precipitation issue in late 2022 with
back processed data available in the NEON 2024 data release. In the meantime, we manually evaluated
the mean annual precipitation recorded at each NEON site against other observational data networks and
noted locations where this issue is generating unexpectedly high or low precipitation values (Table S2).



Another example of how NCAR modeling improved NEON data quality comes from unusual soil
moisture profiles that were initially generated in preliminary simulations at the ABBY site (data not
shown). Upon closer inspection these patterns were found to be caused by an unusual relationship
between soil organic carbon content and depth at this site, which did not match related data gathered
during sample collection or subsequent analyses. Further investigation confirmed that the labels for the
soil carbon analysis subsamples had been switched for two ABBY soil horizons. The NEON soil data
have since been corrected and the modeled soil moisture profiles for ABBY now follow a more typical
pattern with surface soils drying out during the summer and less variation in soil moisture in deeper soil
horizons (Figs. 8, S3). There are also important differences in vertical profiles of simulated and measured
soil moisture, with soil moisture simulated by CTSM typically decreasing with depth while NEON soil
moisture observations generally increase with depth. Additional investigation is needed to determine if
these discrepancies extend to other sites and indicate issues with CTSM simulations or NEON data
products, but it does underscore a synergy in NCAR modeling and NEON measurements that deserves
more attention moving forward.
We see clear opportunities for NEON observations to help guide future model improvements,
especially related to potential biases in phenology (discussed above), photosynthesis (Fig. 7), and other
processes. Some biases in modeled processes are already documented; for example, Wozniak (2020)
found that CTSM underestimates maximum rates of simulated GPP compared to EC observations in
deciduous forest sites. This suggests that implementation of the photosynthesis scheme in CTSM has
parametric or structural issues that prevent high rates of GPP from occurring in the model. Auxiliary data
from NEON that are not always available from other EC flux towers, for example foliar chemistry, can be
used to update parameter values and to evaluate correlated model variables and processes. The
opportunities afforded by NEON's EC and auxiliary data to improve the representation of ecological
processes in CTSM will improve modeled carbon fluxes at NEON towers and may also ameliorate biases
in global simulations.
Finally, the NCAR-NEON system can also facilitate model-informed prioritization of future data
collection efforts. Models can quantify the dominant drivers of uncertainty in model parameters as well as
in response to environmental drivers using ensemble-based methods of parameter uncertainty
propagation and variance decomposition (LeBauer et al. 2013). Site-level CTSM simulations could
therefore help future NEON data collection campaigns to target variables that contribute the most to
uncertainty in modeled ecosystem fluxes and ecosystem responses to global change.
***4.2 Opportunities enabled for research***
The NCAR-NEON system enables research opportunities in the ecology, global change, and
Earth system science communities by: (1) Democratizing access to NCAR models that can be
customized to meet researchers' needs; (2) Providing a platform that leverages NEON observational
datasets for site-level model configuration and evaluation across the diverse range of ecosystems



captured in the NEON design; (3) Facilitating reproducible research workflows; and (4) Providing gap-
filled meteorological data and partitioned EC flux data products.
Through CESM-Lab, the NCAR-NEON system provides access to the full model code and
datasets used to run CTSM on any computing system. This means that researchers are not limited to
NEON locations or to the default configuration of CTSM, nor do they need access to large-scale
computing resources. The CTSM code can be modified and compiled within the container, so researchers
who wish to run simulations with new model parameterizations or with additional model features may now
do so from any computer. Most personal laptop computers are more than sufficient for running site level
simulations, even when using more computationally complex versions of the land model that include, for
example, ecological dynamics (using the Functionally Assembled Terrestrial Ecosystem Simulator,
FATES; Koven et al. 2020) or representative hillslope hydrology (Swenson et al. 2019). Advanced users
can run CTSM at any single point site by making their own input files. Additionally, researchers can
quantify the impact of adjusting model parameters and processes on terrestrial ecosystems under
historical and future climate scenarios. This flexibility is useful for calibrating the model to improve model
performance at a given site, as well as for gaining mechanistic insights into how different processes and
uncertainties affect ecosystem functioning. Broadening access to CTSM also allows researchers to
rapidly compare model output to their own observational datasets, or to existing NEON observational
datasets that are not yet integrated into the NCAR-NEON system.
Moving forward, we see additional NEON data products as providing valuable insights to the
NCAR-NEON system. These could include NEON measurements that are used both as model inputs
(foliar chemistry, phenology and LAI, and historical land use legacies) and as model validation datasets
(including snow depth, vertical profiles of canopy temperature, leaf water potential, litterfall rates, biomass
and vegetation structure, and depth profiles of soil moisture, temperature, carbon and nitrogen). Although
these data have not yet been integrated into the NCAR-NEON system, we are optimistic that existing
tools can help facilitate their integration into research opportunities. We see powerful opportunities to
expand on this approach to integrate information from NEON's Airborne Observation Platform (AOP) into
workflows that extend model capabilities beyond the relatively small footprint of the EC towers. For
example, the AOP light detection and ranging (LiDAR) data would provide information to initialize stand
structure that would be helpful for calibrating reduced complexity configurations of the CTSM-FATES
model (Fisher and Koven, 2020).
The NCAR-NEON system also promotes reproducibility of research in alignment with the FAIR
data principles (Wilkinson et al. 2016), addressing an ongoing challenge facing both ecology and
geosciences (Powers and Hampton 2019; Culina et al. 2020; Kinkade and Shepherd 2021). The NCAR-
NEON system makes it easy for researchers to share their research workflow as part of their publications,
including accompanying code and data. The containerized system also reduces the time required to
configure and run other researchers' workflows, thereby facilitating the process of reproducing previous
studies and expanding existing workflows to answer new research questions.



In addition to enabling opportunities for research with NCAR models, the NCAR-NEON system
also facilitates access to NEON data which can be used for observationally based research or research
using other models. For example, the gap-filled micrometeorological data and partitioned flux data
products provided in the NCAR-NEON system could be used in other projects related to ecological
forecasting and model evaluation that focuses on ecological processes and land model simulations (Best
et al. 2015; Collier et al. 2018; Eyring et al. 2019; Lewis et al. 2022). As latencies in publishing NEON
data are reduced, we intend to provide updated input and evaluation data to the NCAR-NEON system to
enable near-real time hindcasts of ecosystem states and fluxes. In short, we see the information that is
being generated through this activity as a resource to meet data-requirements of the broader Earth
system science community.
**_4.3 Opportunities enabled for teaching_**
The NCAR-NEON system makes it easy to run and visualize site-level simulations that can be
integrated into classroom settings. The NEON Observatory design provides a unique opportunity for
students to access data from world class field research sites and instrumentation in a variety of
ecosystems. Here we highlight two capacities in which this tool can be integrated into classroom
activities. The first is an interactive web-based visualization tool (Table 2). This tool does not require any
software or data downloads, allowing students to access and explore NEON and CTSM data without
running any simulations. Students can explore and compare observational and simulated data for
numerous fluxes at different temporal scales from 45 terrestrial NEON sites (Table S1). Classroom
modules can be developed to probe various ecological questions, including comparisons across sites,
how fluxes change seasonally, and quantification of interannual variability. Instructors can also use this
tool to highlight differences between models and observations, helping students to better understand how
we measure, simulate, and predict ecosystem processes.
A second opportunity for classroom activities is to run simulations using the NCAR-NEON system
within the CESM-Lab container. The flexible cyberinfrastructure, short simulation run times (typically less
than 10 minutes), and simplified coding requirements facilitate running simulations for classroom
applications. Technical challenges are minimal and can be reduced by using a computer lab with Docker
pre-installed and computers that have sufficient memory and space requirements for data downloads, or
by using larger-scale computing resources like university clusters or cloud computing resources. Once
access to the containerized computing environment is established, students can use the available
tutorials to run NEON tower simulations at the site of their choice and evaluate simulated fluxes against
observations (Table 2).
The NCAR-NEON system is flexible, allowing instructors to easily make additional customizations
for their classes. As an example, this cyberinfrastructure tool was used in a graduate level Land-Climate
Interactions Course at Auburn University in the 2021-2022 academic year. First, students performed
CTSM simulations for the Talladega National Forest site (TALL), the NEON site closest to Auburn





University, and compared latent heat flux simulated by CTSM with the NEON measurements using
system tutorials. Next, students were divided into two project groups focusing on either TALL or Ordway-
Swisher Biological Station (OSBS) sites to conduct parameter perturbation experiments using a tutorial
developed by the instructor. Students collected the relevant parameter values from the literature, updated
model parameter files, and performed ten CTSM simulations at each site, finding that GPP was more
sensitive to the selected parameters than latent heat fluxes. These classroom exercises were paired with
a visit to the TALL site to enrich student's experiences and motivate them to design their own
investigation and experiments. Exposure to the NCAR-NEON system has motivated graduate students to
contribute analyses, tutorials, and additional resources to the broader community. For example, one
graduate student compared NEON precipitation measurements with nearby NOAA sites, helping to
identify potentially problematic NEON sensors (Section 4.1), while another is developing a model for
estimating aboveground biomass using ground-based NEON data and remote sensing measurements
(Narine et al. 2020). These examples highlight how the NCAR-NEON system is inspiring the next
generation of scientists.

## Conclusion

Deeper engagement of diverse scientific communities, removing technical barriers, and
increasing access to research data and tools is critical to advance Earth system science, prediction, and
understanding of ecosystem responses to global change. By developing cyberinfrastructure tools that
facilitate the easy and rapid use of measurements, models, and computing tools, the NCAR-NEON
system aims to enable this convergence of climate and ecological sciences and facilitates the
development and testing of data-driven and model-enabled scientific hypotheses. The system provides a
computationally simplified platform for scientific discovery and for rigorous evaluation and improvement of
model simulations and observational data at NEON tower sites. A particular strength of this system is the
auxiliary data collected by the NEON network that is used to inform site-specific model inputs and model
evaluation. With some effort, users can adapt this system to incorporate and simulate flux towers at other
research sites using the 'Processing NEON data' tools linked in Table 2 to guide data formatting. Thus,
future work could expand this system to include gap-filled flux data from other regional and global
networks like AmeriFlux and FLUXNET, allowing for broader spatial coverage. By facilitating community
engagement in modeling and observing terrestrial ecosystems, cyberinfrastructure tools like this are a key
component for building a more intellectually diverse workforce for global change research and Earth
system science.

## Code and Data availability

Datasets created as part of this project are available as a NEON prototype dataset and archived at
NCAR's Geoscience Data Exchange (GDEX) https://doi.org/10.5065/tmmj-sj66. CTSM code is available



through the CTSM github page and archived at https://doi.org/10.5281/zenodo.7342803. Post processing
scripts that used to make figures in this manuscript are available at:
https://github.com/NCAR/neon_scripts.

**Author Contributions**

All authors contributed to writing and review of the software and manuscript. GBB and MSC contributed to
funding acquisition. DLL, WRW, NS, GBB, DD, DL, and MSC contributed to conceptualization and data
curation. DLL, WRW, NS, and DD contributed to formal analysis, software development, validation, and
visualization.

**Competing Interests**

The authors have no competing interests to declare.

**Acknowledgements**

Significant contributions to this work were made by Jim Edwards, Brian Dobbins, Erik Kluzek, and Cove
Sturtevant. This material is based upon work supported by the National Center for Atmospheric
Research, which is a major facility sponsored by the National Science Foundation (NSF) under
Cooperative Agreement No. 1852977 with additional support from NSF award number 2039932. The
National Ecological Observatory Network is a program sponsored by the NSF and operated under
cooperative agreement by Battelle. WRW was also supported by NSF award numbers 1926413,
2031238, and 2120804. SK's contributions were supported by the USDA NIFA grant 2020-67021-32476.
KMD's contributions were supported by NSF award numbers 1702379 and 2044818 and the USDA NIFA,
Hatch project 1025001.



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
