# Peer review of "Overcoming barriers to enable convergence research by integrating ecological and climate sciences: The NCAR-NEON system Version 1"

_EGUsphere, 2023_

## Author Response (AR1)

*We are very grateful for the reviewer comments, which have significantly improved our manuscript. We have incorporated all the requested changes. Of note, we:*

1) *Updated Tables 2 and 3*
2) *Incorporated information about other networks, data resources, and training that are complementary to this work*

*Our responses to individual comments are included in italics below. Please let us know if you have additional questions or require clarification on any of our responses. We look forward to hearing from you.*

**RC1: 'Comment on egusphere-2023-271', Manuel Acosta, 13 Jun 2023**
Review to Manuscript Number: EGUsphere 2023-271

General comments:

The manuscript "Overcoming barriers to enable convergence research by integrating ecological and climate sciences: The NCAR-NEON system - Version 1" mainly reports the results of a computational platform that synergy meteorological data and site-level ecosystem characterizations from the National Ecological Observatory Network (NEON) with the Community Terrestrial System Model that was developed at the National Center for Atmospheric Research (NCAR), simplifying user interface that facilitates access to and use of NEON observations and NCAR models. The information brought up by the authors regarding the created computational platform is significant concerning the issues that are connected with the usability, operability and interpretation of acquired data at the ecosystem level and application of it to models or simulations in Earth system research. In general, I like the overall work, the combination of robust ecological data and the use of them to improve models to create multiple outputs. Moreover, and I found it very fruitful to improve data accessibility, facilitate research to scientists and provide data for educational opportunities.

Specific comments:

1) I suggest improving the information regarding the parameters measured by soil sensor assemblies in Line 121, similar to what was done in the meteorology part (lines 119-120).

*AR: Thanks for this suggestion. In our revised text we clarify that soil sensor assemblies measure depth-resolved soil temperature and moisture at several locations in the EC flux tower footprint.*

2) I consider that the use of the term "Surface characteristics of soil properties and vegetation" (line 124) is not completely correct. I suggest using "Characteristics of soil properties and surface vegetation", which is clear and concise. The same is in Table 1. Data Product Name:

Soil physical and chemical properties, Megapit. It should be identified as Soil Characterization under the Data Product Use, instead of Surface characterization.

*AR: We have updated the text in line 124 to read "Vegetation and soil properties" and updated the description of the megapit data in Table 1 to "Soil property characterization" to more accurately describe the data products.*

3) The use of any acronym must be stated even though is well known. In line 168 is stated netCDF, which stands for Network Common Data Format.

*AR: We updated the text in line 168 to "Network Common Data Form (netCDF)" to define the netCDF acronym.*

4) In Table 2. Four of the seven "URL" links mentioned in the Table are not working. Even though the authors stated that "Note we intend to provide permanent urls for these sites in the final published manuscript", the links are not found. These are Tutorial, Interactive, Processing NEON data and NEON Prototype Data. I was disappointed I could not check the mentioned links and their operability. I hope the links will be active soon.

*AR: We apologize for the inconvenience. All the links were active and working in our documents when we submitted the manuscript, but it looks like the conversion to PDF inserted line breaks that made the links inaccessible. We have updated Table 2 so that the links do not have line breaks, and also updated the links to their final homes. Before final publication we will work with editorial staff and GMD to ensure all links work as intended.*

*Table 2. List of helpful websites created for the NCAR-NEON system, their contents and a url address for each.*

| Name | Contents |
|---|---|
| Project Home Page | Main landing page for users interested in learning more about the project |
| URL: https://neoncollab.ucar.edu | |

| | |
|---|---|
| *Tutorial* | *Tutorial that introduces running CTSM at NEON tower sites in the CESM-Lab container.* |
| *URL: https://ncar.github.io/ncar-neon-books/notebooks/NEON_Simulation_Tutorial.html* | |
| *Interactive Visualizations* | *Interactive plots that allow users to explore data produced by the NCAR-NEON system without running the model or downloading data.* |
| *URL: https://ncar.nationalsciencedatafabric.org/neon-demo/v1/* | |
| *Processing NEON data* | *Docker image with scripts used for gap filling meteorological data, flux partitioning, and formatting NEON datasets.* |
| *URL: https://quay.io/repository/ddurden/ncar-neon* | |
| *DiscussCESM Forum* | *Discussion forum bulletin boards for questions related to CESM including CESM-Lab and CTSM.* |
| *URL: https://bb.cgd.ucar.edu/cesm/* | |
| *CTSM Repository* | *Code base, technical documentation and information related to CTSM* |
| *URL: https://github.com/ESCOMP/CTSM* | |
| *NEON Prototype Data* | *NEON prototype datasets, which include the gap filled meteorological data for flux partitioned data used for model input and evaluations* |
| *URL: https://data.neonscience.org/prototype-datasets/0a56e076-401e-2e0b-97d2-f986e9264a30* | |

5) I suggest changing the name of the subchapter 2.1.2. to characteristics of soil properties and surface vegetation.

*AR: We updated section 2.1.2 to "Soil and vegetation properties" to improve accuracy.*

6) Perhaps will be good to bring a short explication for the statement "equilibrated" carbon, water, energy, and nitrogen states and fluxes, in line 263.

*AR: This sentence was expanded to explain that "For each NEON site, this includes a surface dataset that reflects soil properties and the dominant vegetation (Table 1), meteorological data that*

*provide boundary conditions for the land model, and an initial conditions file with equilibrated, or steady-state, carbon, water, energy, and nitrogen states to initialize ecosystem pools simulated by CTSM"*

7) Again, the use of any acronym must be stated even though is well known. In line 362 is stated Data I/O, which stands for Data Input - Output.

*AR: We removed the acronym in this section, and the updated text reads "Data input-output".*

8) In line 399 is stated "DEJU" that stands for? The official name of the site.

*AR: The DEJU site is Delta Junction in Alaska. We updated the text to state "...for a boreal forest site at Delta Junction (DEJU) in central Alaska)."*

9) It seems is a mistake in the interpretation of Figure 3b-c. In lines, 407-408 is stated "At BART we see that CTSM tends to underestimate sensible heat fluxes, while overestimating latent heat fluxes, especially during the summer months (Fig. 3b-c). The graphs showed different trends compared to those stated in the text.

*AR: Thank you for pointing this out. Indeed, the sensible heat fluxes are overestimated by CTSM while the latent heat fluxes are underestimated by CTSM at the BART site. We updated the text to correct this mistake.*

10) I suggest deleting the word "Democratizing", is too constitutional, perhaps use "Improving".

*AR: We removed 'Democraztizing' and instead state that this system "...enables research opportunities .... by: 1)Facilitating access to NCAR models…"*

Technical corrections

1) in line 201, a closing bracket is missing.

*AR: We updated the text to add a closing bracket.*

**RC2: 'Comment on egusphere-2023-271', Anonymous Referee #2, 01 Jul 2023**

I read with interest the manuscript titled "Overcoming barriers to enable convergence research by integrating ecological and climate sciences: The NCAR-NEON system Version 1" and feel like it makes a timely and relevant contribution to the scientific literature. The manuscript was really well written too, which made it easy to read and understand the science being presented. The specific comments I provide below should help clarify a couple of minor items.

Specific comments:

Line 68: I would have thought appropriate references here would be Novick et al. (2018) and Beringer et al. (2022), which describe Ameriflux and OzFlux, respectively.

*AR: Thank you for the suggestions. We added these references to the text.*

Line 77-79: There are also numerous agricultural sites either in regional flux networks, or not affiliated at all, that represent a barrier for ESM development/parameterization, and ESM use. The authors might consider broadening their discipline scope here

*AR: Thank you for this important point. We have updated the text here to also highlight that researchers in agroecology also face these barriers.*

Table 2: Nice table, and good to see the authors are already thinking about url permanency for the final m/s version

*AR: Thank you. All the links were active and working in our documents during submission, but it looks like the conversion to PDF inserted line breaks that made the links inaccessible. We have updated Table 2 (see updated table in our response to Reviewer 1's comment) so that the links do not have line breaks, and also updated the links to their final homes. Before final publication we will work with editorial staff and GMD to ensure all links work as intended.*

Lines 190-193: Yes, because this approach would not work so well for ecosystems where two plant functional types exist, such as tropical savannas that make up a large portion (20-25 % depending on definition) of the terrestrial land surface. Perhaps the authors could comment on when the CTSM is likely to include more than one PFT

*AR: We agree that a single PFT is likely not representative of each NEON site, and we use this as a starting point. We have added text to highlight that it is possible for users to update the surface dataset to include additional PFTs, and that future work may provide datasets that include multiple PFTs. We note here that this will depend on funding and community needs. The updated text states: "CTSM represents mixed species communities as separate patches occupied by single PFTs. CTSM can represent more than one PFT at each site, and users can update the provided CTSM surface dataset to include more than one PFT and future efforts may provide datasets with multiple PFTs corresponding to their proportions at NEON sites. "*

Line 205: Is this method appropriate across all NEON sites? I realise its out of the scope for this manuscript, but would be somthing to consider whether the daytime approach is more applicable at some sites. Also, do any of the sites have a profile system? Can the authors comment on $CO_2$ storage at low $u^*$ and what effect this might have on gap filling and flux uncertainty across sites?

*AR: Thank you for raising this point. We agree that it's important to evaluate which methods are most appropriate at particular sites and conditions, including storage terms, but such considerations are outside the scope of this manuscript. We have added the following text to Section 2.1.3: "The nighttime approach is a community standard and was used at all sites in this work, and future work can explore whether other partitioning approaches may be more appropriate at some sites." We also add: " In future releases of the NCAR-NEON system we aim to use the ONEFlux data pipeline to harmonize processing with FLUXNET standard data processing (Pastorello et al. 2020)."*

Table 3: I know it can make tables messy, but adding a value for some level of variance/range/uncertainty would be useful for MAT, MAP and GPP to give the reader an idea of annual variability in these parameters. Can ET also be included? Would be useful to see alongside MAP

*AR: Our analysis now includes annual means +/- one standard deviation and includes latent heat flux. These changes have been added to Table 3 (below) and will also be included in SI Table 2 of the revised manuscript*

> **Table 3** *Summary of site name, location, mean annual temperature (MAT), mean annual precipitation (MAP), gross primary production (GPP) and latent heat flux at a subset of NEON sites. Values show annual means and standard deviations in parentheses. Due to gaps in the NEON observational estimates, mean annual GPP and latent heat fluxes are for the full time series simulated by CTSM at each site. All results are for 2018-2021 unless noted otherwise. The full list of results is shown in Tables S1, S2.*

| NEON Site ID | Site Name | Lat | Lon | MAT (C) | MAP (mm y$^{-1}$) | GPP (gC m$^{-2}$ y$^{-1}$) | Latent Heat (W m$^{-2}$) |
|---|---|---|---|---|---|---|---|
| BART | Bartlett Experimental Forest | 44.065 | -71.2883 | 7.7 (0.7) | 1213 (146) | 1126 (57) | 33.6 (1.3) |
| HARV | Harvard Forest | 42.536 | -72.1756 | 8.5 (0.6) | 1404 (502) | 1153 (53) | 32.3 (1.8) |
| STEI | Steigerwaldt-Chequamegon | 45.508 | -89.5888 | 5.7 (0.9) | 659 (110) | 1109 (88) | 29.7 (0.8) |
| KONZ | Konza Prairie Biological Station | 39.101 | -96.5623 | 12.9 (0.7) | 617 (168) | 1158 (235) | 49 (4.8) |
| SRER | Santa Rita Experimental Range | 31.911 | -110.835 | 20.4 (0.7) | 328 (104) | 360 (133) | 26.1 (6.8) |
| ABBY | Abby Road | 45.762 | -122.33 | 10.1 (0.4) | 2042 (409) | 1906 (35) | 29.5 (1.3) |

Line 429: Typo, either due to or related to

*AR: We have corrected this typo*

Line 499: Ah, good to see storage mentioned here. While this does somewhat address my earlier comment, I do think storage should be briefly mentioned in the methods too.

*AR: Thank you for this suggestion. We updated our methods section 2.1.3. It now outlines that NEE is the sum of the turbulent and storage fluxes: "The vapor pressure deficit (VPD) is derived from the difference between actual and saturated vapor pressure, while gross primary production (GPP) is calculated from net ecosystem exchange (NEE), the sum of the turbulent and storage fluxes, using the nighttime flux partitioning method of Reichstein et al. (2005)."*

Lines 655-658: Talking of flux forecasting, the authors fail to mention project EDDIE, funded by the NSF. I believe a few papers have been published relating to this project (see: https://serc.carleton.edu/eddie/about/publications_presentations.html), and suggest the authors cite one or two here to be inclusive.

*AR: Thank you for pointing us to this resource. We agree and "hope that the data and visualizations provided by the NCAR-NEON system and can be integrated with learning modules for undergraduate and graduate training, such as those from Project EDDIE (e.g., Carey et al. 2020, O'Reilly et al 2017), to broaden exposure to large datasets, ecological modeling, and systems thinking."   The quoted text is included in section 4 of our revised manuscript.*

Line 622 (end of section): The NCAR-NEON tool seems like an excellent tool, especially for processing NEON data. However, at this point I'm left wondering how useful this tool might be for sites and users not in NEON, as there hasn't been much discussion about third-party non-NEON sites. There has also been little mention of other EC data processing tools that are already available, such as PyFluxPro used by the OzFlux community or commercial products such as TOVI from LICOR. Indeed, the ONEFLUX processing tool was designed with the purpose of standardising flux processing at a global scale. So, I'm left wondering what place the NCAR-NEON system has amongst these other options. Is it to seamlessly process NEON data quickly or can it be used more widely by other non-NEON sites? I think a little more discussion on these points would be helpful in this section.

*AR: Thanks for this encouragement to expand on these points. In short, the NEON data processing and gap-filling pipeline that is developed here uses redundant data streams measured at the NEON tower sites. NEON is working with Ameriflux to incorporate NEON's redundant data streams into the ONEFlux data processing. We revised the text in Section 4.2 to state: "...and (4) Providing gap-filled meteorological data and partitioned EC flux data products that create synergies with other flux networks and data pipelines (Novick et al. 2018, Beringer et al. 2020; Pastorello et al. 2020).*

*In building the NCAR-NEON system we improved the software infrastructure and workflows that are required to run single point simulations with CTSM, while developing derived, prototype datasets with NEON's EC measurements. Although the focus of this work is on connecting CTSM and NEON data, measurements from non-NEON sites can also be used with this system, facilitating the use of data from additional EC towers and the ONEFlux data pipeline in CTSM development and evaluation. Moving forward, NEON is working with Ameriflux to incorporate the redundant data stream gap-filling from NCAR-NEON with ONEFlux standardized data processing as well as providing proper data formats and metadata for modeling framework ingestion."*

*We also add: "A strength of this system is the auxiliary data collected by the NEON network that is used to inform site-specific model inputs and model evaluation. With some effort, users can adapt this system to incorporate and simulate flux towers at other research sites using the 'Processing NEON data' tools linked in Table 2 to guide data formatting. Thus, future work could expand this system to include gap-filled flux data from other regional and global networks like AmeriFlux and FLUXNET, allowing for broader spatial coverage."*

Line 714-716: Ahh, here's the first mention of how another network could use the NCAR-NEON system, and its in the conclusion. This supports my point from earlier that some discussion around this is needed before this final point. And there are other networks beyond the US that

could benefit from this, such as those that are still establishing or do not have the level of data management sophistication like NEON, OzFlux and ICOS.

*AR: We agree that this is an important point to make earlier in the manuscript. Please see our response to the point above, which we include in section 4.2 of the revised manuscript as well as references cited below. We also include new text at the start of section 4 to further highlight how this project builds upon and contributes to other efforts in providing accessible data, resources, and training.*

References: There seems to be a lot of self-citing in this reference list, i.e. 7 Bonan first author papers. While it is somewhat understandable given the topic and focus of the manuscript, the authors could be more inclusive of other networks and groups who have already made significant progress towards observational data-model integration and in standardising EC processing protocols. By being more inclusive with citations throughout, the manuscript will be more relevant for a global audience and more likely to be cited widely.

*AR: Thanks for this suggestion. Our intent was not to be exclusive, and we appreciate the suggested references. We have added references to highlight other networks and resources (OzFlux, AmeriFlux, EDDIE, and the Land Sites Platform) throughout the manuscript to illustrate that there are several efforts to make ecosystem data accessible and usable.*

*References Added:*

*Beringer, J., Moore, C. E., Cleverly, J., Campbell, D. I., Cleugh, H., De Kauwe, M. G., Kirschbaum, M. U. F., Griebel, A., Grover, S., Huete, A., Hutley, L. B., Laubach, J., van Niel, T., Arndt, S. K., Bennett, A. C., Cernusak, L. A., Eamus, D., Ewenz, C. M., Goodrich, J. P., … Woodgate, W. (2022). Bridge to the future: Important lessons from 20 years of ecosystem observations made by the OzFlux network. Global Change Biology, 28, 3489– 3514.* *https://doi.org/10.1111/gcb.16141*

*Carey, CC, Farrell, KJ, Hounshell, AG, O'Connell, K. Macrosystems EDDIE teaching modules significantly increase ecology students' proficiency and confidence working with ecosystem models and use of systems thinking. Ecol Evol. 2020; 10: 12515– 12527.* *https://doi.org/10.1002/ece3.6757*

*Keetz, L. T., Lieungh, E., Karimi-Asli, K., Geange, S. R., Gelati, E., Tang, H., et al. (2023). Climate–ecosystem modelling made easy: The Land Sites Platform. Global Change Biology, 29, 4440–4452.* *https://doi.org/10.1111/gcb.16808*

*Novick, K. A., Biederman, J. A., Desai, A. R., Litvak, M. E., Moore, D. J. P., Scott, R. L., & Torn, M. S. (2018). The AmeriFlux network: A coalition of the willing. Agricultural And Forest Meteorology, 249, 444-456. doi:* *https://doi.org/10.1016/j.agrformet.2017.10.009.*

*O'Reilly, C.M., Gougis, R.D., Klug, J.L., Carey, C.C., Richardson, D.C., Bader, N.E., Soule, D.C., Castendyk, D., Meixner T., Stromberg, J., Weathers, K.C., and W. Hunter. 2017. Using large data sets for open-ended inquiry in undergraduate science classrooms. Bioscience 67:12:1052-1061. doi.org/10.1093/biosci/bix118*